

# Similar and divergent responses to salinity stress of jamun (*Syzygium cumini* L. Skeels) genotypes

Anshuman Singh[1,2], Ashwani Kumar[1], Jai Prakash[3] and
Arvind Kumar Verma[4]

[1] ICAR–Central Soil Salinity Research Institute, Karnal, Haryana, India
[2] ICAR–Central Institute for Subtropical Horticulture, Lucknow, Uttar Pradesh, India
[3] Division of Fruits and Horticultural Technology, ICAR–IARI, New Delhi, India
[4] ICAR–National Research Centre on Seed Spices, Ajmer, India

Corresponding author
Anshuman Singh, anshuman.
singh@icar.gov.in

## ABSTRACT

**Background:** Genetic variation for salt tolerance remains elusive in jamun (*Syzygium cumini*).
**Methods:** Effects of gradually increased salinity (2.0–12.0 dS/m) were examined in 20 monoembryonic and 28 polyembryonic genotypes of jamun. Six genotypes were additionally assessed for understanding salt-induced changes in gas exchange attributes and antioxidant enzymes.
**Results:** Salt-induced reductions in leaf, stem, root and plant dry mass (PDM) were relatively greater in mono- than in poly-embryonic types. Reductions in PDM relative to control implied more adverse impacts of salinity on genotypes CSJ-28, CSJ-31, CSJ-43 and CSJ-47 (mono) and CSJ-1, CSJ-24, CSJ-26 and CSJ-27 (poly). Comparably, some mono- (CSJ-5, CSJ-18) and poly-embryonic (CSJ-7, CSJ-8, CSJ-14, CSJ-19) genotypes exhibited least reductions in PDM following salt treatment. Most polyembryonic genotypes showed lower reductions in root than in shoot mass, indicating that they may be more adept at absorbing water and nutrients when exposed to salt. The majority of genotypes did not exhibit leaf tip burn and marginal scorch despite significant increases in Na$^+$ and Cl$^-$, suggesting that tissue tolerance existed for storing excess Na$^+$ and Cl$^-$ in vacuoles. Jamun genotypes were likely more efficient in Cl$^-$ exclusion because leaf, stem and root Cl$^-$ levels were consistently lower than those of Na$^+$ under salt treatment. Leaf K$^+$ was particularly little affected in genotypes with high leaf Na$^+$. Lack of discernible differences in leaf, stem and root Ca$^{2+}$ and Mg$^{2+}$ contents between control and salt treatments was likely due to their preferential uptake. Correlation analysis suggested that Na$^+$ probably had a greater inhibitory effect on biomass in both mono- and poly-embryonic types. Discriminant analysis revealed that while stem and root Cl$^-$ probably accounted for shared responses, root Na$^+$, leaf K$^+$ and leaf Cl$^-$ explained divergent responses to salt stress of mono- and poly-embryonic types. Genotypes CSJ-18 and CSJ-19 seemed efficient in fending off oxidative damage caused by salt because of their stronger antioxidant defences.
**Conclusions:** Polyembryonic genotypes CSJ-7, CSJ-8, CSJ-14 and CSJ-19, which showed least reductions in biomass even after prolonged exposure to salinity stress, may be used as salt-tolerant rootstocks. The biochemical and molecular

underpinnings of tissue tolerance to excess $Na^+$ and $Cl^-$ as well as preferential uptake of $K^+$, $Ca^{2+}$, and $Mg^{2+}$ need to be elucidated.

# INTRODUCTION

Salinity stress adversely affects most fruit crops (*Denaxa et al., 2022*; *Niu, Davis & Masabni, 2019*) which abruptly begin to decline due to cumulative impacts of osmotic and ion-specific stresses (*Meir et al., 2014*). While some fruit crops thrive under saline conditions, prolonged exposure to salt stress eventually suppresses their growth and yield (*Rohit Katuri, Trifonov & Arye, 2019*). Salt-induced lowering of soil water potential initially impairs root's ability for water uptake, which then decreases the turgor pressure and hinders shoot and root growth (*Zhao et al., 2020*). Salt-stressed plants accumulate organic and inorganic solutes for osmotic adjustment (*Ahmed et al., 2012*). Stomatal closure brought on by osmotic stress diminishes the plant's capacity to assimilate $CO_2$ (*Chartzoulakis, 2011*). Ionic imbalance, caused by the build-up of $Na^+$ and $Cl^-$, disrupts the metabolic processes by interfering with the activities of essential ions such as $NO_3^-$ and $K^+$ (*Zhao et al., 2020*). This eventually leads to the destruction of cell membranes, reduced activity of enzymes, decreased levels of chlorophyll pigments, and suppression of photosynthesis (*Singh & Sharma, 2018*). Reactive oxygen species (ROS) such superoxides and singlet oxygen build-up abnormally under saline conditions, leading to oxidative stress and lipid peroxidation (*Singh et al., 2023*). Plants activate their antioxidant defense to offset the negative effects of ROS (*Moradbeygi et al., 2020*).

Although intra-specific variation has greatly aided to the selection of salt-tolerant cultivars, the genetic diversity for salt tolerance within many species still remains underutilized (*Ismail & Horie, 2017*). Fruit crops exhibit distinct intra-specific variations when exposed to salt (*Denaxa et al., 2022*; *Etehadpour et al., 2020*). Identifying such genetic differences may lead to the use of salt-tolerant genotypes as rootstock for salt-affected soils. Comprehending the morpho-physiological mechanisms underlying salinity tolerance is also crucial for the development of appropriate breeding techniques aimed at enhancing salt tolerance (*Ismail & Horie, 2017*). Salt-induced changes in biomass allocation and ion uptake differ greatly among genotypes, and serve as valuable indicators for selecting the salt-tolerant cultivars (*Kchaou et al., 2010*). Genetic differences are known for salt-induced declines in biomass (*Pandey et al., 2014*; *Kchaou et al., 2010*). While salt stress typically restricts the availability of essential nutrients (*e.g.*, $K^+$ and $Ca^{2+}$), their levels are little affected in certain genotypes (*Mahouachi, 2018*; *Papadakis et al., 2018*), probably because of preferential uptake (*Liu et al., 2020*).

Jamun (*Syzygium cumini* Skeels) is widely distributed in many Asian and African countries (*Singh et al., 2019*; *Tewari, Singh & Nainwal, 2017*). Various parts of the jamun tree are used as constituents in traditional medicine. Jamun fruit pulp and seed contain a variety of bioactive substances with strong anti-glycaemic, anti-oxidant and

chemoprotective properties (*Kapoor, Ranote & Sharma, 2015*; *Shrikanta, Kumar & Govindaswamy, 2015*). Genetic variations for salt tolerance in Jamun remain elusive, notwithstanding some anecdotal evidence suggesting that this species flourishes under saline conditions (*Sarvade et al., 2016*; *Tewari, Singh & Nainwal, 2017*). This is because the prior salinity experiments in Jamun did not even examine the salt-induced changes in biomass allocation and ion partitioning (*Chhabra & Kumar, 2008*; *Tomar et al., 2003*). Jamun exhibits polyembryony, characterized by the emergence of more than one seedling from a single seed (*Sivasubramaniam & Selvarani, 2012*). Despite the fact that polyembryonic saplings may perform better under salt stress (*Nimbolkar et al., 2019*), the comparative reactions to salt stress of mono- and poly-embryonic seedlings of jamun are not known.

Considering the previously highlighted gaps in research, this experiment was conducted to assess the effects of incremental rise in salinity on biomass allocation and ion partitioning in 20 monoembryonic and 28 polyembryonic genotypes of jamun with the aim of identifying major traits that underpin salt tolerance. The polyembryonic seedlings were distinguished from the monoembryonic types on the basis growth habit *i.e.*, the presence of multiple seedlings per seed. The shared and contrasting responses to salt of mono- and poly-embryonic seedlings were also examined. Six randomly chosen genotypes were additionally assessed to elucidate salt-induced changes in gas exchange attributes and antioxidant enzymes.

## MATERIALS AND METHODS

### Study site and experimental conditions

The experiment was conducted at ICAR-Central Soil Salinity Research Institute, Karnal, India (29°42′31.7″N 76°57′12.7″E) between December 2018 and June 2019. The study area experiences a subtropical climate, with scorching summers, dry winters and total annual rainfall of about 700 mm. Forty-eight (48) genotypes of jamun, comprising both mono- (20) and poly-embryonic (28) types, were examined (Table 1). Approximately six-month-old seedlings were moved to clay pots (upper diameter: 20 cm, basal diameter: 12 cm, height: 18 cm) having 8 kg soil, river sand and farmyard manure (1: 1: 1 v/v) on December 7, 2019. The drainage hole at the bottom of pots was covered with glass wool to prevent the potting soil from leaking out. The experimental plants were kept in a net-house, open on all sides but covered with a polyethylene sheet to prevent the intrusion of rainwater. The experimental pots were reshuffled every 2 weeks to minimize the spatial effects.

### Salinity treatments

Assuming that an incremental rise in salinity would more closely resemble soil salinity under field conditions, where plants rarely encounter sudden surge in salt stress (*Shavrukov, 2013*), salinity of irrigation water was gradually raised every week, beginning at 2 dS/m and increasing to 12 dS/m over time. Salt treatments were imposed on December 21, 2018, 2 weeks after planting. Saline groundwater (electrical conductivity ~15.0–16.0 dS/m) was diluted with fresh water to obtain waters of varying electrical conductivities (2, 4, 6, 8, 10 and 12 dS/m). Fresh water (0.70 dS/m) was used to irrigate the control plants. Table S1

**Table 1 Basic details of jamun genotypes used in the study.**

| S N | Geographic region | State | District | Code | Type* |
|---|---|---|---|---|---|
| 1 | IP | Punjab | Fazilka | CSJ-1 | P |
| 2 | IP | Punjab | Fazilka | CSJ-2 | M |
| 3 | IP | Haryana | Jind | CSJ-3 | P |
| 4 | IP | Haryana | Jind | CSJ-4 | M |
| 5 | IP | Haryana | Jind | CSJ-5 | M |
| 6 | IP | Haryana | Kaithal | CSJ-6 | P |
| 7 | IP | Haryana | Kaithal | CSJ-7 | P |
| 8 | IP | Haryana | Sonipat | CSJ-8 | P |
| 9 | IP | Haryana | Sonipat | CSJ-9 | P |
| 10 | IP | Haryana | Sonipat | CSJ-10 | P |
| 11 | IP | Haryana | Sonipat | CSJ-11 | P |
| 12 | IP | Haryana | Karnal | CSJ-12 | M |
| 13 | IP | Rajasthan | Alwar | CSJ-13 | P |
| 14 | IP | Rajasthan | Ajmer | CSJ-14 | P |
| 15 | IP | Rajasthan | Ajmer | CSJ-15 | M |
| 16 | IP | Rajasthan | Ajmer | CSJ-16 | M |
| 17 | IP | Rajasthan | Pali | CSJ-17 | P |
| 18 | IP | Rajasthan | Rajsamand | CSJ-18 | M |
| 19 | IP | Rajasthan | Sirohi | CSJ-19 | P |
| 20 | GP | Uttarakhand | Dehradun | CSJ-20 | M |
| 21 | GP | Uttar Pradesh | Shamli | CSJ-21 | M |
| 22 | GP | Uttar Pradesh | Muzaffarnagar | CSJ-22 | P |
| 23 | GP | Uttar Pradesh | Aligarh | CSJ-23 | P |
| 24 | GP | Uttar Pradesh | Fatehpur | CSJ-24 | P |
| 25 | GP | Uttar Pradesh | Mirzapur | CSJ-25 | P |
| 26 | GP | Uttar Pradesh | Jaunpur | CSJ-26 | P |
| 27 | GP | Uttar Pradesh | Jaunpur | CSJ-27 | P |
| 28 | GP | Uttar Pradesh | Varanasi | CSJ-28 | M |
| 29 | GP | Bihar | Patna | CSJ-29 | M |
| 30 | GP | Bihar | Patna | CSJ-30 | P |
| 31 | GP | Madhya Pradesh | Gwalior | CSJ-31 | M |
| 32 | EPR | Jharkhand | Gumla | CSJ-32 | M |
| 33 | EPR | Jharkhand | East Singhbhum | CSJ-33 | M |
| 34 | EPR | Chhattisgarh | Bilaspur | CSJ-34 | P |
| 35 | EPR | Odisha | Dhenkanal | CSJ-35 | M |
| 36 | EPR | Telangana | Hyderabad | CSJ-36 | P |
| 37 | EPR | Telangana | Medak | CSJ-37 | P |
| 38 | EPR | Assam | Jorhat | CSJ-38 | M |
| 39 | WPR | Gujarat | Bharuch | CSJ-39 | P |
| 40 | WPR | Gujarat | Bharuch | CSJ-40 | P |
| 41 | WPR | Gujarat | Bhavnagar | CSJ-41 | P |

| S N | Geographic region | State | District | Code | Type* |
|-----|-------------------|-------|----------|------|-------|
| | **Table 1** (continued) | | | | |
| 42 | WPR | Gujarat | Bhavnagar | CSJ-42 | M |
| 43 | WPR | Gujarat | Bhavnagar | CSJ-43 | M |
| 44 | WPR | Gujarat | Bhavnagar | CSJ-44 | P |
| 45 | WPR | Maharashtra | Amravati | CSJ-45 | M |
| 46 | WPR | Maharashtra | Nandurbar | CSJ-46 | P |
| 47 | WPR | Maharashtra | Nandurbar | CSJ-47 | M |
| 48 | WPR | Karnataka | Haveri | CSJ-48 | P |

**Note:**
IP, Indus plains; GP, Gangetic plains; EP, Eastern peninsular region; WPR, Western peninsular region; *Seedling type: M, monoembryonic; P, polyembryonic.

shows the compositions of fresh and saline waters. Irrigation water was poured in using a graduated beaker until it evenly reached the bottom of the pots. The plants were watered until May 31 2019, when treatments were stopped with the appearance of leaf injury symptoms in some mono- (CSJ-28, CSJ-31) and poly-embryonic (CSJ-1, CSJ-34) genotypes.

## Observations recorded

### Leaf, stem and root biomass

The observations were recorded 5 weeks after salinity level of 12.0 dS/m was imposed. The plants were gently uprooted, cleaned on a wire screen, and briefly shade dried. Then, the individual plants were separated into leaves (excluding 4th pair from apex), stems and roots, washed using distilled water, put within envelopes, and oven-dried to a constant weight (NSW, Gurugram, India). Leaf, stem and root dry mass were recorded using an electronic balance. Plant dry mass was determined by adding the mass of each component.

### Gas exchange attributes and anti-oxidant enzymes

The mature leaves (4th pair from apex) were tagged for recording the photosynthetic attributes and anti-oxidant enzymes. Six genotypes (CSJ-1, CSJ-13, CSJ-18, CSJ-19, CSJ-28 and CSJ-42) showing noticeably distinct responses to salinity stress were randomly selected to assess the salt-induced changes in gas exchange attributes and anti-oxidant enzymes. Net photosynthesis ($P_n$), transpiration rate ($E$) and internal $CO_2$ concentration ($C_i$) were measured using a portable photosynthesis system (6400 XT; LI-COR, Lincoln, NE, USA) (*Singh et al., 2024*). The ratio between $P_n$ and $E$ was used to compute the instantaneous water usage efficiency (WUE). Ascorbate peroxidase (APX) and superoxide dismutase (SOD) activities were determined using the procedures described in *Nakano & Asada (1981)* and *Beauchamp & Fridovich (1971)*, respectively. The methodologies given in *Aebi (1984)* and *Rao et al. (1998)* were adopted to measure catalase (CAT) and peroxidase (POX) activities, respectively.

*Mineral ions*

For ion analyses, finely ground leaf (4[th] pair from apex), stem and root tissues (50 mg each) were used. A flame photometer (Systronics, Ahmedabad, India) was used to measure $Na^+$ and $K^+$, an ion-selective electrode (Eutech, Singapore) for determining $Cl^-$, and an atomic absorption spectrometer (Analytik Jena, Jena, Germany) for measuring $Ca^{2+}$ and $Mg^{2+}$ contents.

## Statistical analyses

A randomized block design with four replications was used. The independent and interaction effects of salinity (fixed factor) and genotype (random factor) on the variance in different traits were assessed by a two-way analysis of variance (*Doncaster & Davey, 2007*). The comparative reactions of mono- and poly-embryonic types to fresh and salt water treatments were examined by Welch's *t*-test (JASP v. 0.17.3). Welch's test is considered to be more appropriate when sample sizes are unequal. The strength and directionality of associations between the measured traits were determined by computing the Pearson's bivariate correlations. Linear discriminant analysis (LDA) was used for discerning the shared and contrasting responses to salinity stress. A confusion matrix was generated to predict the group membership from LDA (PAST v. 4.10).

## RESULTS

### Biomass allocation

Salinity stress reduced leaf (LDM), stem (SDM), root (RDM) and plant (PDM) dry mass by 24.76, 33.16, 25.16 and 26.55%, respectively, in monoembryonic types, and by 21.92, 30.73, 18.46 and 22.61%, respectively, in polyembryonic types ($p < 0.001$) (Table 2). While CSJ-28 showed the greatest decreases in LDM (41.77%), SDM (58.42%) and RDM (48.61%), CSJ-29 and CSJ-31 (LDM), CSJ-2, CSJ-15, CSJ-31 and CSJ-47 (SDM), and CSJ-12, CSJ-33, CSJ-43 and CSJ-47 (RDM) were other monoembryonic genotypes that showed noticeable declines (>35.0%) in biomass attributes when treated with salt. Comparably, only minor reductions (<15%) in LDM (CSJ-5, CSJ-18, CSJ-45), SDM (CSJ-5, CSJ-18, CSJ-35), and RDM (CSJ-16, CSJ-18, CSJ-21) were seen in some genotypes. Within polyembryonic types, salt-induced reductions in LDM were notable (>30.0%) in CSJ-22, CSJ-34, CSJ-44 and CSJ-46; and the least in CSJ-19 (8.46%) and CSJ-40 (2.47%). The decreases in SDM ranged between 2.40% (CSJ-17) and 58.48% (CSJ-1), and in RDM from 1.95% (CSJ-44) to 36.76% (CSJ-26). The highest (36.28%) and the lowest (11.02%) decreases in PDM under salinity stress were seen in CSJ-27 and CSJ-19, respectively (Table S2). Shoot: root ratio (SRR) declined by 4.30 and 7.72% in mono- and poly-embryonic types, respectively, under salt treatment. However, the decrease was significant ($p = 0.005$) only in the latter (Table 2). Only a few genotypes (mono: CSJ-16, CSJ-31; poly: CSJ-23, CSJ-44, CSJ-46) showed modest decreases (>25.0%) in SRR when treated with salt. Interestingly, SSR also appreciably increased in some genotypes including CSJ-12 and CSJ-43 (mono) and CSJ-40 (poly) (Table S2).

**Table 2 Effects of salinity, genotype and their interaction on biomass traits in monoembryonic and polyembryonic types.**

| Trait | C Monoembryonic | S | Change (%) | Source | $p$ | C Polyembryonic | S | Change (%) | Source | $p$ |
|---|---|---|---|---|---|---|---|---|---|---|
| LDM | 19.06 | 14.34 | −24.76 | S | <0.001 | 21.17 | 16.53 | −21.92 | S | <0.001 |
| | | | | G | 0.008 | | | | G | <0.001 |
| | | | | S × G | <0.001 | | | | S × G | <0.001 |
| SDM | 7.48 | 5 | −33.16 | S | <0.001 | 7.68 | 5.32 | −30.73 | S | <0.001 |
| | | | | G | <0.001 | | | | G | <0.001 |
| | | | | S × G | <0.001 | | | | S × G | <0.001 |
| RDM | 10.65 | 7.97 | −25.16 | S | <0.001 | 11.54 | 9.41 | −18.46 | S | <0.001 |
| | | | | G | <0.001 | | | | G | <0.001 |
| | | | | S × G | <0.001 | | | | S × G | <0.001 |
| PDM | 37.18 | 27.31 | −26.55 | S | <0.001 | 40.39 | 31.26 | −22.61 | S | <0.001 |
| | | | | G | <0.001 | | | | G | <0.001 |
| | | | | S × G | <0.001 | | | | S × G | <0.001 |
| SRR | 2.79 | 2.67 | −4.30 | S | 0.235ns | 2.59 | 2.39 | −7.72 | S | 0.005 |
| | | | | G | <0.001 | | | | G | <0.001 |
| | | | | S × G | <0.001 | | | | S × G | <0.001 |

**Note:**
LDM, leaf dry mass (g/plant); SDM, stem dry mass (g/plant); RDM, root dry mass (g/plant); PDM, plant dry mass (g/plant); SRR, shoot to root ratio. C and S denote control (0.70 dS/m) and salt treatments (12.0 dS/m). S, salinity, G, genotype. ns, non-significant ($p > 0.05$).

## Leaf ions

Under salt treatment, leaf $Na^+$ and $Cl^-$ were approximately 150.0 and 44.0% higher in monoembryonic types, and 184.0 and 53.0% higher in polyembryonic types, respectively. While leaf $K^+$ appreciably decreased (~20.0–22.0%) and leaf $Mg^{2+}$ slightly increased in salt-stressed plants (6.0–7.0%), leaf $Ca^{2+}$ did not vary significantly between control and salt treatments, regardless of the seedling type (Table 3). Monoembryonic genotypes CSJ-45 (411.70%) and CSJ-43 (53.26%) showed the highest and lowest increases in leaf $Na^+$ than respective controls. Genotypes CSJ-21, CSJ-32, CSJ-33, CSJ-38 and CSJ-47 had over threefold higher leaf $Na^+$ under saline conditions. Despite overall significant reductions, only CSJ-2, CSJ-21, CSJ-29 and CSJ-38 experienced appreciable declines (>30.0%) in leaf $K^+$ when treated with salt. Salt-induced increases in leaf $Cl^-$ ranged between 1.98% (CSJ-45) and 134.25% (CSJ-20). Apart from CSJ-45, in which leaf $Ca^{2+}$ declined appreciably (58.0%), salt-triggered declines or increases in leaf $Ca^{2+}$ were rather small (~15.0%) in other genotypes. Likewise, only CSJ-21 (33.12%) and CSJ-43 (36.93%) showed appreciable increases in leaf $Mg^{2+}$ under the salt treatment (Table S3). In polyembryonic types, salt-induced upticks in leaf $Na^+$ ranged between 18.83% (CSJ-3) and 518.97% (CSJ-41). Genotypes CSJ-22 (437.50%), CSJ-23 (506.25%), CSJ-24 (341.38%) and CSJ-48 (354.0%) also showed substantial increases in leaf $Na^+$ under saline conditions. Increases in leaf $Cl^-$ ranged from 4.03% (CSJ-6) to 233.93% (CSJ-26). With notable exceptions of CSJ-8 (96.58%), CSJ-10 (203.28%), CSJ-36 (111.54%) and CSJ-48 (98.08%), salt-triggered increases in leaf $Cl^-$ in most other genotypes were <50.0% compared to controls (Table S3). Of polyembryonic genotypes, only CSJ-34 (44.23%), CSJ-41 (58.51%) and CSJ-44 (57.53%)

**Table 3 Effects of salinity, genotype and their interaction on leaf mineral ions in monoembryonic and polyembryonic types.**

| Trait | C Monoembryonic | S | Change (%) | Source | $p$ | C Polyembryonic | S | Change (%) | Source | $p$ |
|---|---|---|---|---|---|---|---|---|---|---|
| $Na^+$ | 1.40 | 3.52 | +151.43 | S | <0.001 | 1.21 | 3.43 | +183.47 | S | <0.001 |
| | | | | G | <0.001 | | | | G | 0.020 |
| | | | | S × G | <0.001 | | | | S × G | <0.001 |
| $K^+$ | 4.04 | 3.21 | −20.55 | S | <0.001 | 4.64 | 3.61 | 22.19 | S | <0.001 |
| | | | | G | 0.014 | | | | G | <0.001 |
| | | | | S × G | <0.001 | | | | S × G | <0.001 |
| $Cl^-$ | 0.82 | 1.18 | +43.90 | S | <0.001 | 0.93 | 1.42 | +52.69 | S | <0.001 |
| | | | | G | <0.001 | | | | G | <0.001 |
| | | | | S × G | <0.001 | | | | S × G | <0.001 |
| $Ca^{2+}$ | 17.01 | 15.48 | −8.99 | S | 0.079ns | 19.44 | 19.23 | −1.08 | S | 0.862ns |
| | | | | G | <0.001 | | | | G | <0.001 |
| | | | | S × G | <0.001 | | | | S × G | <0.001 |
| $Mg^{2+}$ | 7.67 | 8.16 | +6.39 | S | 0.020 | 7.45 | 7.96 | +6.85 | S | 0.002 |
| | | | | G | <0.001 | | | | G | <0.001 |
| | | | | S × G | <0.001 | | | | S × G | <0.001 |

**Note:**
C and S denote control (0.70 dS/m) and salt treatments (12.0 dS/m). All ion contents are in mg/g DW, S- salinity, G- genotype. ns- non-significant ($p > 0.05$).

showed appreciable declines in leaf $K^+$ following salt treatment. In response to salt, leaf $Ca^{2+}$ and $Mg^{2+}$ either rose, decreased, or remained identical. For instance, leaf $Ca^{2+}$ modestly decreased (>20.0%) in CSJ-3, CSJ-8 and CSJ-48, slightly increased (>20.0%) in CSJ-23, CSJ-24, CSJ-40 and CSJ-44, and essentially remained unchanged in CSJ-1, CSJ-7, CSJ-14, CSJ-25, CSJ-37 and CSJ-39. Similarly, with the exceptions of CSJ-3 (18.05%), CSJ-8 (10.78%), CSJ-9 (5.82%), CSJ-27 (6.54%) and CSJ-48 (5.35) in which leaf $Mg^{2+}$ slightly dropped, and of CSJ-7, CSJ-11, CSJ-19 and CSJ-36 in which it did not change, leaf $Mg^{2+}$ increased in other genotypes under the salt treatment (Table S3).

### Stem ions

While stem $Na^+$ and $Cl^-$ significantly increased (>50.0%), stem $K^+$ significantly decreased (≥20.0%), regardless of seedling type, in response to salt. Although stem $Mg^{2+}$ was not significantly different between control and salt treatments, stem $Ca^{2+}$ significantly increased only in polyembryonic types (Table 4). In monoembryonic types, salt-triggered increases in stem $Na^+$ were either marked (282.0% in CSJ-32), modest (~100.0% in CSJ-2, CSJ-5 and CSJ-45) or hardly different from controls (CSJ-12 and CSJ-31). Apart from CSJ-12 and CSJ-15, stem $K^+$ declined in other genotypes under saline conditions. Nonetheless, declines in stem $K^+$ were noticeable (>50.0%) only in CSJ-42 and CSJ-43. The highest (202.23%) and lowest (1.43%) increases in stem $Cl^-$ than control were seen in CSJ-35 and CSJ-16, respectively (Table S3). Stem $Ca^{2+}$ and $Mg^{2+}$ increased or decreased in salt-stressed plants. Only two genotypes (CSJ-28 and CSJ-31) exhibited appreciable (>50.0%) decreases in stem $Ca^{2+}$ under salt stress. In response to salt, stem $Mg^{2+}$ appreciably rose (>40.0%; CSJ-15, CSJ-16 and CSJ-33) or dropped (40.0–60.0%; CSJ-31 and CSJ-43) in some

**Table 4 Effects of salinity, genotype and their interaction on stem ion contents in monoembryonic and polyembryonic types.**

| Trait | C Monoembryonic | S | Change (%) | Source | $p$ | C Polyembryonic | S | Change (%) | Source | $p$ |
|---|---|---|---|---|---|---|---|---|---|---|
| $Na^+$ | 1.69 | 2.66 | +57.39 | S | <0.001 | 1.83 | 2.76 | +50.82 | S | <0.001 |
| | | | | G | 0.700 ns | | | | G | 0.003 |
| | | | | S × G | <0.001 | | | | S × G | <0.001 |
| $K^+$ | 8.95 | 6.50 | −27.37 | S | <0.001 | 8.78 | 7.08 | −19.36 | S | <0.001 |
| | | | | G | 0.049 | | | | G | 0.014 |
| | | | | S × G | <0.001 | | | | S × G | <0.001 |
| $Cl^-$ | 0.48 | 0.73 | +52.08 | S | <0.001 | 0.50 | 0.76 | +52.0 | S | <0.001 |
| | | | | G | <0.001 | | | | G | <0.001 |
| | | | | S × G | <0.001 | | | | S × G | <0.001 |
| $Ca^{2+}$ | 11.28 | 12.22 | +8.33 | S | 0.323 ns | 11.05 | 13.88 | +25.61 | S | 0.039 |
| | | | | G | 0.129 ns | | | | G | 0.044 |
| | | | | S × G | <0.001 | | | | S × G | <0.001 |
| $Mg^{2+}$ | 5.42 | 5.65 | +4.24 | S | 0.952 ns | 5.50 | 5.83 | +6.0 | S | 0.275 ns |
| | | | | G | <0.001 | | | | G | 0.001 |
| | | | | S × G | <0.001 | | | | S × G | <0.001 |

Note:
C and S denote control (0.70 dS/m) and salt treatments (12.0 dS/m). All ion contents are in mg/g DW, S- salinity, G- genotype. ns- non-significant ($p > 0.05$).

genotypes, but was little affected in others (Table S3). Of the polyembryonic genotypes, CSJ-1 and CSJ-7 displayed the greatest (159.21%) and the smallest (1.11%) increases in stem $Na^+$ when exposed to salt. Salt-triggered increases in stem $Cl^-$ varied from 1.52% in CSJ-19 to 249.15% in CSJ-27. Salinity stress caused noticeable upticks in stem $Cl^-$ (≥150.0%) only in genotypes CSJ-6, CSJ-37 and CSJ-40 (Table S3). Except for a few genotypes (CSJ-3, CSJ-10, CSJ-27 and CSJ-40), where it rose over control, stem $K^+$ dropped under salinity stress in other genotypes. Depending on genotype, stem $Ca^{2+}$ either increased [1.91% (CSJ-1) to 274.32% (CSJ-40)] or decreased [5.05% (CSJ-48) to 43.96% (CSJ-11)] under the salt treatment. The response of stem $Mg^{2+}$ to salt was also genotype-specific, increasing [1.75% (CSJ-24) to 81.03% (CSJ-40)], decreasing [0.71% (CSJ-9) to 52.01% (CSJ-11)], or remaining unchanged (CSJ-1) (Table S3).

## Root ions

In monoembryonic types, salt-induced increases in root $Na^+$ and $Cl^-$ were quite similar (~39.0%); however, $K^+$, $Ca^{2+}$ and $Mg^{2+}$ contents were not significantly affected. In polyembryonic types, root $Na^+$, $Cl^-$ and $Ca^{2+}$ increased by 51.52, 30.77 and 17.79%, respectively; however, there was no discernible difference between control and salt treatments for root $K^+$ and $Mg^{2+}$ (Table 5). Monoembryonic genotypes CSJ-21 and CSJ-32 exhibited the greatest (115.13%) and the smallest (1.05%) increases, respectively, in root $Na^+$ relative to controls. Genotypes CSJ-2, CSJ-28, CSJ-33 and CSJ-42 showed only negligible increases (<15.0%) in root $Na^+$ under salt stress. Salt-induced increases in root $Cl^-$ ranged between 1.27% (CSJ-32) and 369.23% (CSJ-31); CSJ-5, CSJ-15, CSJ-21, CSJ-3, CSJ-35 and CSJ-47 showed only slight increases (≤15.0%) in root $Cl^-$. Salt stress did not

**Table 5 Effects of salinity, genotype and their interaction on root ion contents in monoembryonic and polyembryonic types.**

| Trait | C Monoembryonic | S | Change (%) | Source | $p$ | C Polyembryonic | S | Change (%) | Source | $p$ |
|---|---|---|---|---|---|---|---|---|---|---|
| $Na^+$ | 1.63 | 2.25 | +38.04 | S | <0.001 | 1.65 | 2.50 | +51.52 | S | <0.001 |
| | | | | G | 0.412ns | | | | G | 0.126ns |
| | | | | S × G | <0.001 | | | | S × G | <0.001 |
| $K^+$ | 3.62 | 3.58 | −1.11 | S | 0.742ns | 3.70 | 3.59 | −2.97 | S | 0.299ns |
| | | | | G | 0.161ns | | | | G | 0.002 |
| | | | | S × G | <0.001 | | | | S × G | <0.001 |
| $Cl^-$ | 0.51 | 0.71 | +39.22 | S | <0.001 | 0.52 | 0.68 | +30.77 | S | <0.001 |
| | | | | G | 0.021 | | | | G | <0.001 |
| | | | | S × G | <0.001 | | | | S × G | <0.001 |
| $Ca^{2+}$ | 14.10 | 11.59 | −17.80 | S | 0.162ns | 13.04 | 15.36 | +17.79 | S | 0.020 |
| | | | | G | 0.260ns | | | | G | 0.077ns |
| | | | | S × G | <0.001 | | | | S × G | <0.001 |
| $Mg^{2+}$ | 3.87 | 3.84 | −0.78 | S | 0.836ns | 3.60 | 3.84 | +6.67 | S | 0.302ns |
| | | | | G | 0.219ns | | | | G | 0.038 |
| | | | | S × G | <0.001 | | | | S × G | <0.001 |

Note:
C and S denote control (0.70 dS/m) and salt treatments (12.0 dS/m). All ion contents are in mg/g DW, S- salinity, G- genotype. ns- non-significant ($p > 0.05$).

significantly alter root $K^+$, $Ca^{2+}$ and $Mg^{2+}$ levels (Table S3). In polyembryonic types, while root $Na^+$ remained virtually unchanged in some genotypes (CSJ-44, CSJ-48), certain genotypes (CSJ-44, CSJ-48) exhibited only a modest increase (<20.0%), whereas others (CSJ-1, CSJ-39) displayed appreciable upticks (>100.0%) in root $Na^+$. With a few exceptions, increases in root $Cl^-$ induced by salt were less than increases in root $Na^+$; genotypes CSJ-40 and CSJ-34 exhibited the lowest (2.63%) and the highest (146.34%) upticks in root $Cl^-$ in comparison to controls. Root $K^+$ either increased or decreased in response to salinity; however, only a few genotypes demonstrated appreciable (15.0–20.0%) decreases (CSJ-17, CSJ-23, CSJ-24 and CSJ-41) or increases (CSJ-13, CSJ-37 and CSJ-46) in root $K^+$. Despite either an increase or decrease in root $Ca^{2+}$ under salt treatment, the genotypic differences were not significant. A more or less comparable pattern was also seen for root $Mg^{2+}$ (Table S3).

### Comparative responses under control and saline conditions

The polyembryonic types showed significantly higher LDM ($t$ = 5.77, $p$ = < 0.001) and PDM ($t$ = 3.30, $p$ = 0.001) than monoembryonic types under control treatment. Similarly, they also exhibited significantly higher ($p < 0.001$) LDM, RDM and PDM (16.53, 9.41 and 31.26 g/plant, respectively) than monoembryonic types (14.35, 7.97 and 27.31 g/plant, respectively) when treated with salt. Shoot: root ratio was significantly higher ($p = 0.009$) in monoembryonic types under salt treatment (Table S4; Fig. 1). Of the leaf ions, the two groups differed in $Na^+$ only in control and in Cl only in salt treatment. Comparably, $K^+$ and $Ca^{2+}$ were different under both fresh and salt treatments. Leaf $Mg^+$ did not differ

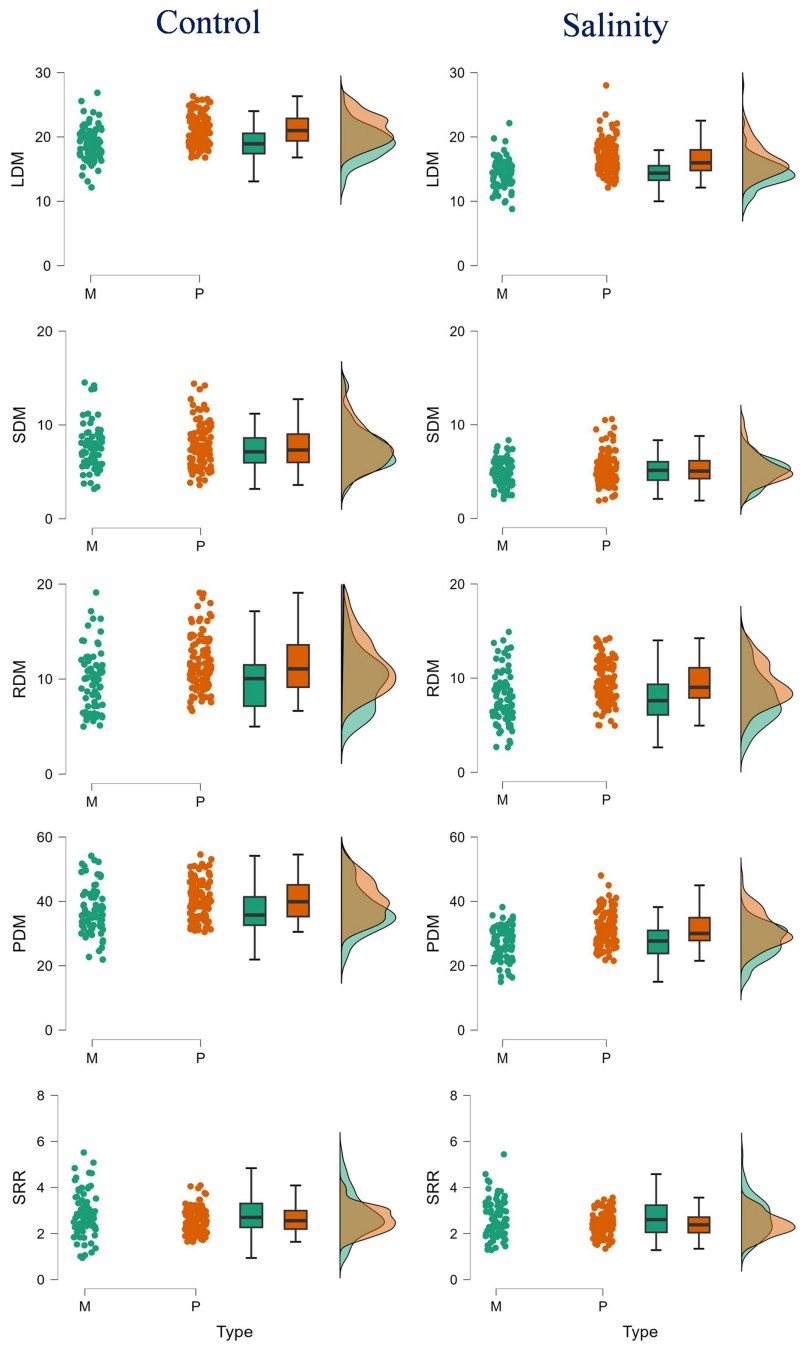

**Figure 1 Raincloud plots showing the differences between monoembryonic and polyembryonic types in biomass traits under control (0.70 dS/m) and salinity (12.0 dS/m) treatments.** LDM, leaf dry mass (g/plant); SDM, stem dry mass (g/plant); RDM, root dry mass (g/plant); PDM, plant dry mass (g/plant); SRR, shoot: root ratio. M and P indicate monoembryonic and polyembryonic types, respectively.

significantly under both the conditions (Fig. 2A). Of the stem ions, differences between mono- and poly-embryonic types for $Na^+$ were significant only in control and for $K^+$ and $Ca^{2+}$ in salt treatment (Fig. 2B). In roots, only $Na^+$ (10.57%) and $Ca^{2+}$ (32.50%) were

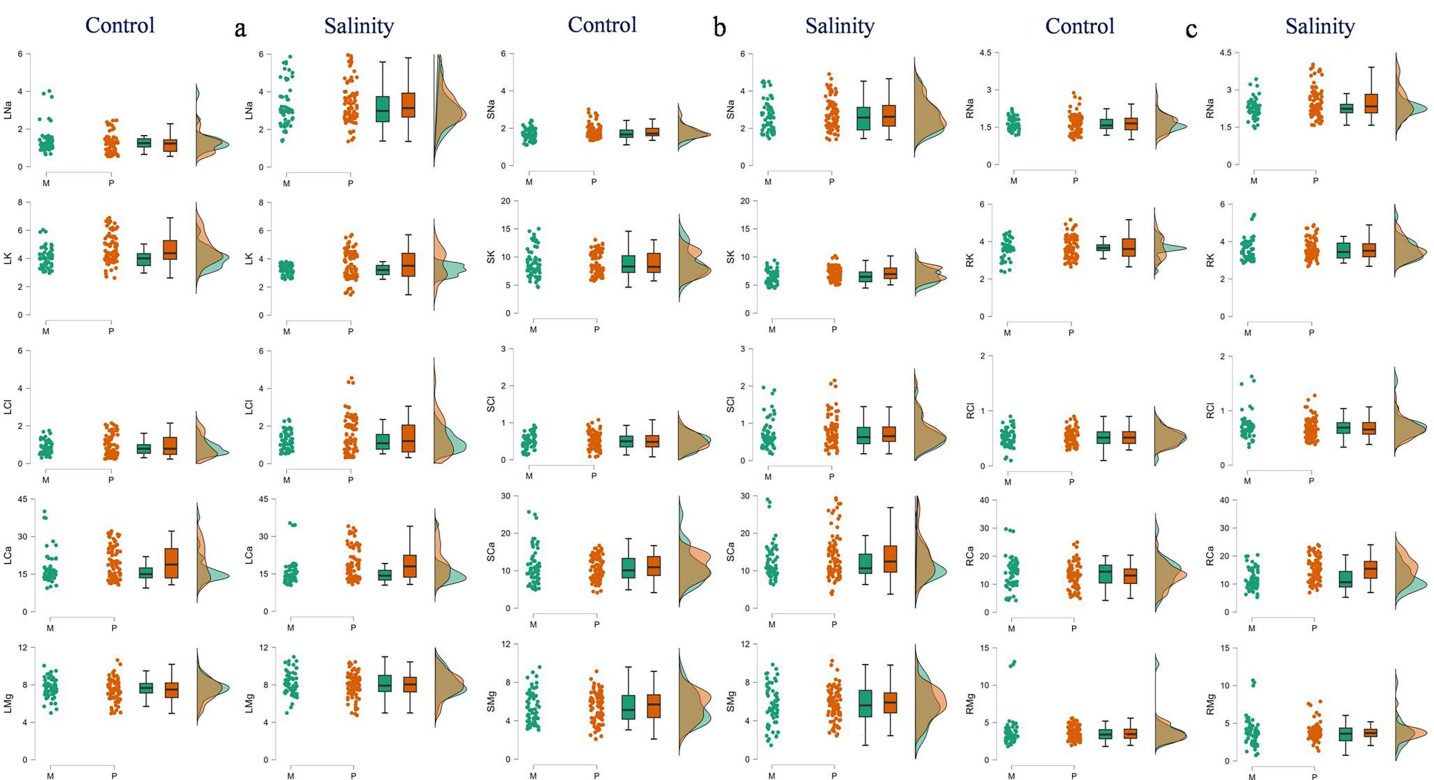

**Figure 2 Raincloud plots showing the differences between monoembryonic and polyembryonic types in leaf (A), stem (B) and root (C) ion concentrations under control (0.70 dS/m) and salinity (12.0 dS/m) treatments.** LNa: leaf Na$^+$, LK–leaf K$^+$, LCl–leaf Cl$^-$, LCa–leaf Ca$^{2+}$, LMg–leaf Mg$^{2+}$, SNa: stem Na$^+$, SK–stem K$^+$, SCl–stem Cl$^-$, SCa–stem Ca$^{2+}$, SMg–stem Mg$^{2+}$, RNa: root Na$^+$, RK–root K$^+$, RCl–root Cl$^-$, RCa–root Ca$^{2+}$, RMg–root Mg$^{2+}$. All ion concentrations are in mg/g DW. M and P indicate monoembryonic and polyembryonic types, respectively.

significantly higher in polyembryonic types under salt treatment; the differences were non-significant for other ions under both control and salt treatments (Table S4; Fig. 2C).

## Correlation analysis

Table S5 shows the Pearson's correlations between the measured traits. LDM, SDM, RDM and PDM were all significantly positively correlated with each other, irrespective of seedling type. Leaf Na$^+$ had highly significant negative correlations with LDM, SDM, RDM and PDM. Although all biomass attributes exhibited significant negative correlations with leaf Cl$^-$, the degree of association was invariably lower than that between biomass traits and leaf Na$^+$. While relationships of leaf K$^+$ and stem Na$^+$ with biomass attributes were respectively significantly positive and significantly negative. Stem Cl$^-$ had significant negative relationships with LDM and PDM in both mono- and poly-embryonic types, and with SDM only in polyembryonic types. Stem K$^+$ had significant positive correlations with all the biomass traits in monoembryonic types, but only with LDM and PDM in the polyembryonic types. The negative correlations between root Na$^+$ and biomass traits were greater in poly- than in mono-embryonic types.

## Linear discriminant analysis

Table S6 and Fig. 3 display the results of LDA. The first two discriminant functions alone explained approximately 98.0% of the cumulative variance in the data, indicating that LDA efficiently reduced the dimensionality. We found that while LD-1 was mainly a construct of stem $Cl^-$ and root $Cl^-$, LD-2 had root $Na^+$, leaf $K^+$ and leaf $Cl^-$ as the highly weighted variables. The differences between mono- and poly-embryonic types for stem and root $Cl^-$ contents were non-significant under both control and salinity treatments. Conversely, the differences between two groups were significant for root $Na^+$, leaf $K^+$, and leaf $Cl^-$ under salt treatment (Table S6). Therefore, we infer that while stem and root $Cl^-$ contents accounted for shared responses, root $Na^+$, leaf $K^+$ and leaf $Cl^-$ explained the divergent responses to salinity stress. A perusal of the LDA biplot showed that while LD1 effectively discriminated the control and salinity treatments, LD2 could fairly reasonably distinguish monoembryonic and polyembryonic types from each other (Fig. 3). The confusion matrix estimates for predicted group membership from LDA are presented in Table S7. The overall classification accuracy (jacknifed) of LDA was 80.21%. In the case of monoembryonic types, 25.0% and 18.75% of the instances were mislabelled as polyembryonic in the control and salt treatments, respectively. Similarly, 19.64% and 13.39% of the polyembryonic instances were incorrectly classified as monoembryonic under control and salinity treatments, respectively. Interestingly, a small proportion of the polyembryonic instances (3.57%) were also incorrectly labelled as monoembryonic in the salt treatment.

## Gas exchange parameters and anti-oxidant enzymes

Six genotypes, comprising both monoembryonic (CSJ-18, CSJ-28 and CSJ-42) and polyembryonic (CSJ-1, CSJ-13 and CSJ-19) types, that exhibited noticeably distinct responses under saline conditions were further examined to assess the effects of salinity stress on photosynthetic attributes and antioxidant enzymes (Table 6). Salt-induced reductions in net photosynthesis ($P_n$) ranged between 20.47% (CSJ-18) and 83.37% (CSJ-1). The highest (36.25%) and the lowest (16.90%) declines in transpiration rate ($E$) were recorded in genotypes CSJ-42 and CSJ-19, respectively. With the exception of CSJ-19 (17.85%), salinity-triggered decreases in internal $CO_2$ concentration ($C_i$) were not much different (36.66–45.26%) in other genotypes. Salt stress suppressed the WUE to varying extents; reductions ranged between 1.89% (CSJ-18) and 76.46% (CSJ-1) (Table 6). The tested genotypes demonstrated substantial differences in the levels of antioxidant enzymes under both control and salinity treatments. Genotypes CSJ-18 (77.81%) and CSJ-19 (73.50%) demonstrated remarkable increases in APX activity when treated with salt; by comparison, APX activity was only about 28.0% higher in salt-stressed CSJ-1 plants. Salt-induced increases in CAT activity ranged between 8.19% (CSJ-1) and 63.48% (CSJ-18) (Table 6). Remarkable genotypic differences were also evident for POX activity; it increased by 12.76, 31.83, 46.67, 45.82, 24.29 and 25.38% in genotypes CSJ-1, CSJ-13, CSJ-18, CSJ-19, CSJ-28 and CSJ-42. Under saline conditions, SOD activity varied remarkably among the genotypes; it was the lowest (26.96%) in CSJ-1 and the highest (70.87%) in CSJ-19 in comparison to salt-free plants (Table 6).

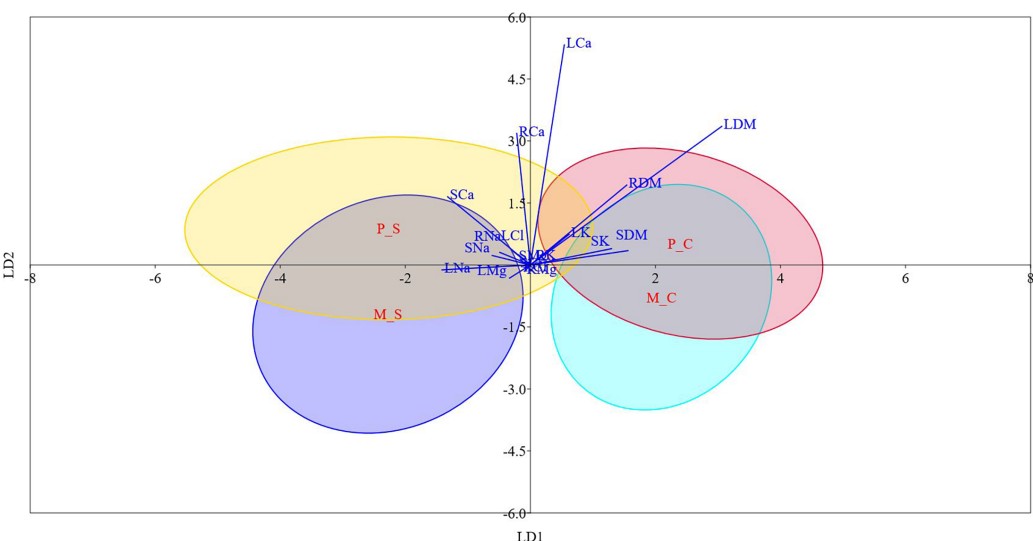

**Figure 3 Linear discriminant analysis biplot showing the grouping of different type-treatment combinations on first two linear discriminant functions.** M_C, monoembryonic control; M_S, monoembryonic saline; P_C, polyembryonic control; P_S, polyembryonic saline.

## DISCUSSION

Little is known about traits underlying salinity tolerance, as well as genetic variations for salt tolerance, in Jamun. In this backdrop, our study aimed to elucidate the effects of salinity stress on biomass allocation and ion uptake in 48 diverse genotypes of jamun. Similar and contrasting responses to salt stress of mono- and poly-embryonic seedlings were also analyzed to identify the mechanisms underlying salinity tolerance. In order to avoid salt shock, irrigation water salinity was increased gradually (*Singh et al., 2024*). Since 'osmotic' rather than 'salinity' stress usually triggers early reactions under saline conditions, long-term studies are probably more dependable for analyzing salt tolerance (*Zhu et al., 2016*). By using this method, we were also able in better mimicking the field conditions where salinity typically peaks in the summer (*Sadder et al., 2021*).The genetic variability for salt tolerance remains elusive in Jamun because prior studies had examined the effects of salt stress only on one genotype (*Chhabra & Kumar, 2008*; *Patil & Patil, 1983*; *Tomar et al., 2003*). Crop genotypes differ markedly in salt tolerance (*Liu et al., 2020*; *Mousavi et al., 2019*), suggesting that a sizeable number of genotypes must be evaluated to reasonably assess the variability for salt tolerance. We noticed substantial genotypic differences ($p < 0.001$) for salt-induced declines in the dry mass of different plant parts. In monoembryonic types, while CSJ-28 was most adversely affected, genotypes CSJ-5, CSJ-18 and CSJ-38 exhibited the lowest drops in leaf, stem and root biomass. Contrasting genetic variation was also observed within polyembryonic types for the reductions caused by salt in LDM, SDM and RDM. Salt stress suppresses the leaf area, damages the cell membranes, impairs the water relations, causes oxidative stress and hampers the photosynthetic assimilation, which then adversely impact plant growth (*Gholami et al., 2023*; *Moula et al., 2020*). The repressive effects of salinity on shoot and root growth vary

**Table 6 Analysis of variance and mean comparisons for testing the effects of salinity (S), genotype (G) and their interaction (S × G) on variance in gas exchange attributes and anti-oxidant enzymes.**

| Genotype | Treatment | $P_n$ | E | $C_i$ | WUE |
|---|---|---|---|---|---|
| CSJ-1 | C | 4.21 ± 0.16a | 0.51 ± 0.03e | 272.58 ± 7.68c | 8.37 ± 0.72a |
| | S | 0.70 ± 0.04i | 0.35 ± 0.02f | 155.64 ± 2.74i | 1.98 ± 0.18fg |
| CSJ-13 | C | 4.43 ± 0.16a | 0.52 ± 0.02de | 366.35 ± 2.82a | 8.63 ± 0.61a |
| | S | 1.26 ± 0.08h | 0.38 ± 0.02f | 209.20 ± 7.03f | 3.34 ± 0.40de |
| CSJ-18 | C | 2.98 ± 0.12c | 0.71 ± 0.03b | 358.35 ± 5.26a | 4.23 ± 0.31c |
| | S | 2.37 ± 0.05e | 0.57 ± 0.03cd | 196.17 ± 2.08g | 4.15 ± 0.27cd |
| CSJ-19 | C | 3.66 ± 0.10b | 0.71 ± 0.03b | 301.72 ± 2.65b | 5.14 ± 0.27b |
| | S | 2.65 ± 0.08d | 0.59 ± 0.02c | 247.87 ± 2.76e | 4.51 ± 0.20bc |
| CSJ-28 | C | 3.12 ± 0.13c | 0.82 ± 0.03a | 258.89 ± 2.48d | 3.81 ± 0.09cd |
| | S | 1.58 ± 0.07g | 0.59 ± 0.04c | 160.11 ± 2.15hi | 2.68 ± 0.26ef |
| CSJ-42 | C | 2.09 ± 0.10f | 0.80 ± 0.03a | 261.91 ± 2.42d | 2.61 ± 0.09ef |
| | S | 0.87 ± 0.04i | 0.51 ± 0.03e | 165.90 ± 2.94h | 1.72 ± 0.14g |
| S | | * | ** | ** | ns |
| G | | ns | ** | ns | ns |
| S × G | | *** | *** | *** | *** |
| Genotype | Treatment | APX | CAT | POX | SOD |
| CSJ-1 | C | 9.89 ± 0.39c | 2.32 ± 0.08de | 21.0 ± 0.63e | 12.87 ± 0.41e |
| | S | 12.64 ± 0.40b | 2.51 ± 0.07d | 23.68 ± 0.57d | 16.34 ± 0.55c |
| CSJ-13 | C | 12.52 ± 0.50b | 2.02 ± 0.07f | 26.96 ± 0.89c | 16.65 ± 0.90c |
| | S | 20.01 ± 0.93a | 2.84 ± 0.06c | 35.54 ± 0.91b | 24.20 ± 0.66b |
| CSJ-18 | C | 7.03 ± 0.20d | 2.30 ± 0.10d-f | 26.74 ± 1.44c | 10.09 ± 0.50f |
| | S | 12.50 ± 0.57b | 3.76 ± 0.20a | 39.22 ± 0.87a | 15.83 ± 0.64cd |
| CSJ-19 | C | 6.0 ± 0.13d | 2.20 ± 0.14ef | 7.90 ± 0.33g | 16.24 ± 0.69c |
| | S | 10.41 ± 0.56c | 3.32 ± 0.08b | 11.52 ± 0.52f | 27.75 ± 1.12a |
| CSJ-28 | C | 6.41 ± 0.38d | 2.27 ± 0.08d-f | 21.12 ± 1.29e | 9.88 ± 0.38f |
| | S | 9.20 ± 0.50c | 2.89 ± 0.11c | 26.25 ± 0.59c | 13.28 ± 0.80e |
| CSJ-42 | C | 12.78 ± 0.43b | 2.95 ± 0.18c | 22.38 ± 1.33de | 10.07 ± 0.96f |
| | S | 21.21 ± 1.14a | 3.99 ± 0.16a | 28.06 ± 0.71c | 14.20 ± 0.48de |
| S | | ** | ** | ** | ** |
| G | | ** | ns | ** | * |
| S × G | | *** | *** | *** | *** |

**Note:**
$P_n$, net photosynthesis (µmol $CO_2$/m²/s); E, transpiration rate (mmol $H_2O$/m²/s); $C_i$, internal $CO_2$ concentration (µmol/mol); WUE, water use efficiency (µmol/mmol); APX, ascrobate peroxidise; CAT, catalase; POX, peroxidise; SOD, superoxide dismutase (all anti-oxidants are in units/g FW). S, salinity; G, genotype; * significant at $p < 0.05$; ** significant at $p < 0.01$; *** significant at $p < 0.001$; ns, non-significant. C and S denote control (0.70 dS/m) and salinity (12.0 dS/m) treatments, respectively. Each value represents mean ± SD. Differences between means that share a letter within each column are not statistically significant ($p$ 0.05).

with genotype in a particular fruit crop (*Liu et al., 2020*; *Moula et al., 2020*). We found that while several monoembryonic genotypes (*e.g.*, CSJ-2, CSJ-18 and CSJ-28) showed comparable reductions, a few (*e.g.*, CSJ-5, CSJ-12 and CSJ-45) showed higher decreases in root mass, and the remaining showed greater declines in shoot mass. Likewise, salt-induced reductions in root mass were either lower or substantially lower than

decreases in shoot mass in most polyembryonic genotypes. Preferential allocation of biomass to shoots or roots may imply genotype-specific adaptations to overcome excessive salt accumulations. Maintenance of root biomass may also improve water and nutrient uptake in saline soils (*Perica, Goreta & Selak, 2008*; *Rasheed, Bakhsh & Qadir, 2020*).

Salinity (10.0–12.0 dS/m) had little effects on $Na^+$, $K^+$, $Ca^{2+}$ and $Mg^{2+}$ contents in jamun plants (*Patil & Patil, 1983*; *Yousaf et al., 2020*). In our study, leaf, stem and root $Na^+$ increased to varying degrees, irrespective of the seedling type, following salt treatment. Salt-induced upticks in leaf $Na^+$ (<100.0%) suggested comparatively efficient $Na^+$ exclusion in some genotypes (mono: CSJ-2, CSJ-5, CSJ-29, CSJ-42, and CSJ-43; poly: CSJ-3, CSJ-6, CSJ-10, CSJ-13 and CSJ-26) (*Brumos et al., 2010*; *Etehadpour et al., 2020*; *Mahouachi, 2018*). Both mono- (20.55 and 27.37%, respectively) and poly-embryonic (22.19 and 19.36%, respectively) types showed relatively minor decreases in leaf, stem and root $K^+$ under salt stress. Leaf $K^+$ was hardly different between control and salinity treatments in some genotypes (mono: CSJ-4, CSJ-5, CSJ-28 and CSJ-32, poly: CSJ-13, CSJ-14, CSJ-22 and CSJ-23). Some mono- (CSJ-16, CSJ-28, CSJ-31, CSJ-35, CSJ-38 and CSJ-45), and poly-embryonic (CSJ-6, CSJ-11, CSJ-14, CSJ-25, CSJ-27 and CSJ-40) genotypes also exhibited strong capacities for $Cl^-$ exclusion from leaves. Such variations in $Na^+$, $K^+$ and $Cl^-$ levels in various plant parts were likely because of distinct ion uptake and partitioning mechanisms (*Gengmao et al., 2015*; *Lovelli et al., 2012*), and implied genotypic differences for ion retention in roots (*Zarei et al., 2016*). Despite significant increases in $Na^+$ and $Cl^-$, lack of salt-specific injuries in most genotypes suggested tissue tolerance to the excess $Na^+$ and $Cl^-$ (*Munns et al., 2016*). Salinity up to 12.0 dS/m did not cause leaf scorch and stem dieback in jamun plants (*Patil & Patil, 1983*). Pomegranate (*Sun et al., 2018*) and sapota (*Kumar et al., 2023*) plants did not exhibit leaf tip burn and marginal scorch even after extended exposure to salinity.

Ion translocation from roots to shoots may also account for genetic differences in salt tolerance (*Khoshbakht, Ramin & Baninasab, 2015*; *Calzone et al., 2020*). Under salinity stress, leaf $Na^+$ was lower than root $Na^+$ in certain genotypes (mono: CSJ-16, CSJ-21, CSJ-42 and CSJ-43; poly: CSJ-3, CSJ-8, CSJ-13, CSJ-14, CSJ-39 and CSJ-40). Similarly, leaf and root $Na^+$ contents were fairly similar in some genotypes (CSJ-1, CSJ-11, CSJ-21, CSJ-22, CSJ-30 and CSJ-39) with relatively higher $Na^+$ (>3.0 mg/g DW) in roots. Leaf $Cl^-$ was also lower than root $Cl^-$ in many genotypes (mono: CSJ-21, CSJ-28, CSJ-32 and CSJ-42; poly: CSJ-22, CSJ-23, CSJ-24, CSJ-25, CSJ-27, CSJ-39 and CSJ-40). Such genotypes seemed to limit $Na^+$ and $Cl^-$ translocation from roots to shoots (*Musacchi, Quartieri & Tagliavini, 2006*). Interestingly, leaf, stem and root $Cl^-$ levels, regardless of the seedling type, were consistently lower than those of $Na^+$ under salt treatment, implying a better efficiency for $Cl^-$ exclusion (*Mehdi-Tounsi et al., 2017*). This may also enhance tolerance to salt since excessive $Cl^-$ commonly causes leaf chlorosis and plant mortality, even when $Na^+$ levels are low (*Liu et al., 2018*). Only a few genotypes displayed noticeable declines in leaf (mono: CSJ-2, CSJ-21, CSJ-29, CSJ-38; poly: CSJ-34, CSJ-41, CSJ-44) and stem (mono: CSJ-42, CSJ-43; poly: CSJ-10, CSJ-11, CSJ-17, CSJ-36, CSJ-37) $K^+$ when treated with salt. Leaf $K^+$ levels under the salt treatment were particularly little affected in genotypes with very high leaf $Na^+$ (*i.e.*, > 4.0 mg/g DW). Sufficient $K^+$ levels could have enabled these genotypes in

sequestering excess $Na^+$ in vacuoles (*Zarei et al., 2016*). It also implied a strong selectivity for $K^+$ uptake over $Na^+$ (*Musacchi, Quartieri & Tagliavini, 2006*; *Teixeira & Carvalho, 2009*). Salt stress did not significantly alter leaf, stem and root $Ca^{2+}$ in the monoembryonic types. Salt-triggered decreases in leaf $Ca^{2+}$ were modest (~15.0%) in most genotypes. Similarly, while leaf $Mg^{2+}$ slightly increased, the differences in stem and root $Mg^{2+}$ were hardly discernible between salt-free and salt-stressed plants. In polyembryonic types, salt stress caused significant upticks only in stem and root $Ca^{2+}$. Numerous crops preferentially accumulate $Ca^{2+}$ and $Mg^{2+}$, or at the very least, maintain their levels when exposed to salt (*Liu et al., 2020*; *Mahouachi, 2018*; *Teixeira & Carvalho, 2009*). Sufficient $Ca^{2+}$ and $Mg^{2+}$ levels improve the osmotic adjustment (*Mahouachi, 2018*). Adequate $Ca^{2+}$ probably also enhances selective uptake of $K^+$ over $Na^+$ (*Gengmao et al., 2015*), and improves cell membrane integrity (*Cimato et al., 2010*). Interestingly, leaf, stem and root $Ca^{2+}$ and $Mg^{2+}$ were substantially greater than $Na^+$ and $Cl^-$ contents, irrespective of treatment and genotype, indicating that jamun plants can overcome nutritional constraints imposed by salt (*Cuiyu et al., 2020*; *Liu et al., 2020*).

When treated with salt, the polyembryonic types had significantly higher LDM, RDM and PDM than monoembryonic types. While the differences in leaf and stem $Na^+$ were not statistically significant, polyembryonic types retained significantly more $Na^+$ in their roots under salinity stress. Comparably, leaf $Cl^-$ was noticeably lower in salt-stressed monoembryonic types. This suggested comparatively greater capacity for $Cl^-$ exclusion in monoembryonic seedlings, and for $Na^+$ exclusion in polyembryonic types (*Dayal et al., 2014*; *Hussain et al., 2012*). This is probably because different mechanisms regulate the absorption and partitioning of $Na^+$ and $Cl^-$ in salt-stressed plants (*Saleh et al., 2008*). Importantly, polyembryonic types maintained significantly higher levels of leaf $K^+$, leaf $Ca^{2+}$, stem $K^+$, stem $Ca^{2+}$, and root $Ca^{2+}$ when exposed to salt. In addition to boosting osmotic adjustment (*Mahouachi, 2018*) and improving cell membrane stability (*Cimato et al., 2010*), this may have also restricted $Na^+$ uptake (*Gengmao et al., 2015*), leading to higher leaf and root biomass in the polyembryonic types. Our results broadly concur with earlier findings in mango (*Pandey et al., 2014*) and citrus (*Hussain et al., 2012*), which suggest that polyembryonic genotypes may perform better under salt stress.

Correlation analysis indicated that $Na^+$ probably had a stronger restrictive effect on leaf, stem and root biomass than $Cl^-$, regardless of the seedling type. The greater inhibitory effects of $Na^+$ on plant growth are known in citrus (*Balal et al., 2012*) and olive (*Perica, Goreta & Selak, 2008*). In our study, salt stress caused increased build-up of $Na^+$ in different plant parts. $Na^+$ transport in plants is primarily unidirectional with little recirculation from shoots to roots, which causes $Na^+$ to gradually build-up in shoots. The higher $Na^+$ levels then cause metabolic toxicity by competing with $K^+$ in cellular functions (*Tester & Davenport, 2003*). Contrarily, phloem recirculation seems to limit $Cl^-$ accumulation in aerial plant parts (*Godfrey et al., 2019*). Thus, we suppose that phloem recirculation might shield jamun plants against $Cl^-$ toxicity (*Brumos et al., 2010*). Significant positive correlations between biomass attributes and leaf $K^+$ implied that enhanced accumulation of $K^+$ may contribute to osmotic adjustment (*Pérez-Pérez et al., 2009*) and facilitate sequestration of excess $Na^+$ into vacuoles (*Zarei et al., 2016*).

Approximately 98.0% of the cumulative variance in the data was described by the first two linear discriminant functions alone, demonstrating the robustness of LDA in reducing the dimensionality (*Ye & Ji, 2010*). Because the variables (stem and root Cl⁻) loaded heavily on first linear discriminant function (LD-1) were not significantly different between both the groups under control and saline conditions, we assume that this discriminant function represented the shared responses to salinity stress. Similarly, we assume that LD-2 represented the divergent responses to salinity stress, since the major variables on LD-2 (root $Na^+$, leaf $K^+$ and leaf $Cl^-$) significantly differed between the two groups under salt treatment. Earlier, the most significant features driving the responses of grape (*Bari et al., 2021*) and olive (*Boshkovski et al., 2022*) genotypes to salt stress were reliably delineated by discriminant analysis. In our study, the overall classification accuracy of LDA was 80.21%, quite similar to the values reported in *Boshkovski et al. (2022)*.

Salinity-triggered declines in $P_n$ varied between 20.47% (CSJ-18) and 83.37% (CSJ-1). Interestingly, the genotypes showing the largest salt-induced decreases in $P_n$ (CSJ-1 and CSJ-13) also had the highest $P_n$ rates in the absence of salt (*Mousavi et al., 2019*). The decreases in $P_n$ also seemed to be largely independent of leaf $Na^+$ and $Cl^-$ contents. For instance, despite quite similar increases in leaf $Na^+$ (109.16 and 97.89%, respectively), genotypes CSJ-18 and CSJ-42 showed remarkably different reductions in $P_n$ (*i.e.*, 20.46 and 58.37%, respectively). Similarly, genotypes CSJ-1 and CSJ-18 showed 83.38 and 20.46% decreases in $P_n$, respectively, despite salt-induced increases in leaf $Cl^-$ being 19.23 and 80.77%, respectively (*Alipour, 2018*; *Hussain et al., 2012*). The tested genotypes showed varying degrees of reductions in transpiration rate ($E$) and internal $CO_2$ ($C_i$). Despite being crucial for regulating ion absorption, reduced transpiration can impede plant growth by lowering the photosynthesis (*Negrão, Schmöckel & Tester, 2017*). The activity of antioxidant enzymes also showed significant genotypic differences. For instance, while APX activity increased markedly (>70.0%) in response to salt in CSJ-18 and CSJ-19, it increased by only ~28.0% in CSJ-1. CSJ-1 also displayed the lowest (~8.0%) increase in CAT activity while it was the highest (63.48%) in CSJ-18. When exposed to salt, genotypes CSJ-18 and CSJ-19 also showed significantly higher levels of POX and SOD than other genotypes. This implies that jamun genotypes react differently in terms of antioxidant enzyme activity to oxidative stress brought on by salt (*Ayaz et al., 2021*; *Singh et al., 2023*). The antioxidant enzymes shield the salt-stressed plants from oxidative damage by detoxifying the ROS and regulating ROS formation (*Moradbeygi et al., 2020*), and are believed to be potential markers for identifying the salt-tolerant genotypes (*Sorkheh et al., 2012*). We noticed that genotypes CSJ-18 and CSJ-19 were particularly efficient in upregulating anti-oxidant enzymes. Certain genotypes are frequently better at fending-off oxidative damage caused by salt because they have stronger antioxidant defenses (*Abid et al., 2020*; *Ayaz et al., 2021*; *Etehadpour et al., 2020*). Genotypic variations in antioxidant activities can be attributed to their intricate expression (*Racchi, 2013*) and organelle-specific activities (*Niu & Liao, 2016*) within plant cells.

## CONCLUSIONS

Our results demonstrated distinct genotypic responses to salt within both mono- and poly-embryonic types. The decreases brought on by salt in leaf, stem, root and whole plant biomass were relatively greater in monoembryonic than in polyembryonic types. Furthermore, most polyembryonic genotypes exhibited lower or much lower reductions in root than in shoot mass when treated with salt, suggesting that they might be more adept at absorbing water and nutrients in saline soils. Despite significant increases in $Na^+$ and $Cl^-$ in different plant parts, leaf tip burn and marginal scorch were not seen in the majority of the genotypes. While this raised the possibility of tissue tolerance, which assists in storing excess $Na^+$ and $Cl^-$ in vacuoles, we also assume that preferential accumulation of $K^+$, $Ca^{2+}$ and $Mg^{2+}$ may have played a role in osmotic adjustment and decreased $Na^+$ uptake. Discriminant analysis suggested that while stem and root $Cl^-$ were likely responsible for the common reactions, root $Na^+$, leaf $K^+$ and leaf $Cl^-$ accounted for the divergent responses to salt of the mono- and poly-embryonic types. Some polyembryonic genotypes (CSJ-7, CSJ-8, CSJ-14 and CSJ-19), found to be least affected by salt treatment, could be used as salt-tolerant rootstocks. All the genotypes evaluated by us, including the promising polyembryonic types which propagate true-to-type from seeds, are being maintained in a salt-affected field for further investigation and use. Future studies should aim at delineating the plausible factors accounting for tissue tolerance to excess $Na^+$ and $Cl^-$, and preferential uptake of $K^+$, $Ca^{2+}$ and $Mg^{2+}$.

## ABBREVIATIONS

| | |
|---|---|
| **APX** | ascorbate peroxidise |
| **CAT** | catalase |
| **$C_i$** | intercellular $CO_2$ concentration |
| **$E$** | transpiration rate |
| **$EC_e$** | soil saturation extract electrical conductivity |
| **LDA** | linear discriminant analysis |
| **LDM** | leaf dry mass |
| **$P_n$** | net photosynthesis |
| **POX** | peroxidase |
| **RDM** | root dry mass |
| **ROS** | reactive oxygen species |
| **SDM** | stem dry mass |
| **SOD** | superoxide dismutase |
| **WUE** | water use efficiency |

## ACKNOWLEDGEMENTS

We thank Rajesh Kumar, Amrik Singh, Satpal Sharma, Ajit Singh, Neeraj Kumar, Rakesh Banyal, Mohan Ram, A. R. Chinchmalatapure, Akash Singh, H. Y. Shewale, A. S. Tomar, Pradeep Singh, Ashok Singh, Satpal, Anoop Singh, D. S. Kar, Ankit Goswami, Arvind Upadhyay and Basant Singh for their help in the collection of jamun genotypes. Director,

ICAR-CSSRI, Karnal is appreciated for logistic support. Dheeraj Kumar and Dinesh Meena are thanked for their technical help.

### Funding

This work was supported financially and logistically by ICAR–CSSRI, Karnal, India. No financial support was received from any external agency. The funders had no role in study design, data collection and analysis, decision to publish, or preparation of the manuscript.

### Grant Disclosures

The following grant information was disclosed by the authors:
ICAR-CSSRI.

### Competing Interests

Anshuman Singh is an Academic Editor for PeerJ.

### Author Contributions

- Anshuman Singh conceived and designed the experiments, performed the experiments, analyzed the data, prepared figures and/or tables, authored or reviewed drafts of the article, and approved the final draft.
- Ashwani Kumar conceived and designed the experiments, performed the experiments, authored or reviewed drafts of the article, and approved the final draft.
- Jai Prakash analyzed the data, prepared figures and/or tables, authored or reviewed drafts of the article, and approved the final draft.
- Arvind Kumar Verma analyzed the data, prepared figures and/or tables, authored or reviewed drafts of the article, and approved the final draft.

### Data Availability

The raw data is available in the Supplemental File.

### Supplemental Information

Supplemental information for this article can be found online at http://dx.doi.org/10.7717/peerj.17311#supplemental-information.

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
