# Peer review of "Similar and divergent responses to salinity stress of jamun (Syzygium cumini L. Skeels) genotypes"

_PeerJ, doi:10.7717/peerj.17311_

## Round 0.1 · original submission · Major Revisions

Authors have presented work on an underutilized fruit crop with significant medicinal benefits. However, Reviewer 1 has a few critical comments, which the authors must take into account while revising the manuscript.

·

Basic reporting

The article is authored in a clearly understandable way. In my thought, the author has presented various aspects of the study in a good manner. All the pertinent information about the experiment such as the study site, experimental material, growth environment, salt stress imposition, and standard methods for data recording is presented in full. Similarly, author has used different data analytics to derive major findings. Overall, the results and discussion parts are well written.

Experimental design

The methodological details outlined for the investigation are clear and adequate. The experiment design (Randomized block design) is appropriate. Data have been properly analyzed.

Validity of the findings

The effects of salinity have been evaluated utilizing a wide range of parameters. This study had used 48 diverse genotypes of jamun to draw inferences. This appears adequate number of germplasm to obtain a fair understanding about salt effects. This was a relatively long experiment, assessing salinity stress impacts on important attributes for plant biomass, mineral absorption and partitioning as well as some key physiological traits. Future studies on salinity tolerance may take key messages and insights from the methods used and findings reported herein.

1. In the background section of Abstract, the sentence may be revised as ‘Genetic variation for salt tolerance remains elusive in jamun (Syzygium cumini)’.
2. In the conclusion part of abstract, it is stated that Polyembryonic genotypes CSJ-7, CSJ-8, CSJ-14 and CSJ-19, which showed least reductions in leaf, root and plant biomass even after prolonged exposure to salinity stress, may particularly be useful as salt-tolerant rootstocks. Here, it is advised to keep only term biomass and not leaf, root and plant biomass. Also, in place of ‘may particularly be useful’, write as ‘may be used as’.
3. In line 67-69 of Introduction, author mentions that ‘Comprehending the morpho-physiological mechanisms underlying salinity tolerance is also crucial for the development of appropriate breeding techniques aimed at enhancing salt tolerance’. However, this is not supported by relevant reference, and therefore should be looked into.
4. In line 85-88 of Introduction, the sentence should be revised for a better meaning. It may be written as ‘Despite the fact that polyembryonic saplings may perform better under salt stress (Nimbolkar et al. 2019), the comparative reactions of mono- and poly-embryonic seedlings of jamun to salt stress are not known.’
5. The sentence written as ‘This study intended to….’ (line 88) may be revised as ‘Considering the previously highlighted gaps in research, this experiment was conducted….’
6. In observation recorded part, write as ‘Leaf, stem and root dry mass were recorded…….’ and not as it is mentioned.
7. It is stated that gas exchange traits were determined by using a gas exchange system. However, I suggest writing it as portable photosynthesis system.
8. I could see that author has discussed the effect of salinity on biomass allocation to shoot and root (line 336-344). I also found that related data for shoot and root dry mass are provided in Supplementary Table 2. However, there is no mention of this in the results section. Therefore, it is advised to present the relevant data in table 2 along with results at the appropriate place in manuscript.
9. In lines 329-332, author has described the genotype differences for reductions in biomass along with numeric values which is essentially a repetition of the results already presented. So, it may be removed.
10. In discussion, the authors have stated that salinity stress increased leaf Na in mono-embryonic and poly-embryonic types to differing extents (line 347-349). This is clearly a repetition of the results presented under the head leaf ions (line 169-174). Therefore, this should be removed in order to avoid repetition.
11. In the conclusions part of manuscript, authors suggest that some poly-embryonic genotypes least affected by salt may be used as salt tolerant rootstocks. This raises an important question whether such genotypes have been conserved and maintained for future applications.

Additional comments

The finding presented in the manuscript ‘Similar and divergent responses to salinity stress of jamun (Syzygium cumini L. Skeels) genotypes: biomass allocation and ion partitioning’ improve the existing knowledge with respect to salt tolerance in jamun which is a neglected crop yet has a lot of therapeutic value. A perusal of manuscript revealed that little is known for salinity tolerance in jamun. In this context, this work is important as author studied a good number of jamun genotypes using biomass, mineral ion and physiological parameters. Another significant point is that authors have compared the responses of two seedling populations (monoembryonic and polyembryonic) to understand how they differ when subjected to salt stress. Although manuscript is mostly in order, I have found some deficiencies and shortcomings that must be rectified before the manuscript is accepted for publication.

·

Basic reporting

The author has presented the different parts of the study in an easy language that is understandable. Details regarding the experiment like plant material, experimental setting, application of salt stress and data recording are properly described. In a similar vein, the author has derived important conclusions using different statistical tools. Result and discussion are worded nicely overall.

Experimental design

The methodological specifics are sufficient and well defined. The design used for the study is correct. I did not find any problem in the way data are analyzed and presented.

Validity of the findings

Salinity's impacts have been assessed using biomass parameters, ion uptake and storage in different parts, besides measurement of physiological traits such as photosynthetic parameters and anti-oxidant defence enzymes. Author evaluated the response of 20 monoembryonic and 28 polyembryonic jamun genotypes under a prolonged salt stress. I think the number of genotypes used is good enough to gain insight into salt tolerance mechanisms. Key lessons from this study may be applied in future research on salinity tolerance in jamun and similar crops.

Additional comments

1. Comparative assessment of salt tolerance of two seedling types (mono vs polyembryonic) is a major aspect of this study. Although authors mention about the existence of polyembryony in jamun citing Sivasubramaniam and Selvarani (2012), they have not mentioned about how seedlings were distinguished based on growth habit. Accordingly, selection of two types of seedlings based on growth habit may be indicated at appropriate place.

2. It has been mentioned that ‘comparative reactions of mono- and poly-embryonic types to fresh and salt water treatments were examined by Welch’s t-test, can authors give reason for using Welch’s test in place of commonly used Student’s t-test.

3. Results are overall clearly written. However, there are some minor grammatical mistakes that should be corrected. And, I would like to mention that at some places, the results may be clubbed together instead of being presented in two separate sentences. For example, the first two sentences under the heading ‘leaf ions’ may be suitably revised and combined into one.

4. I am advising the authors to cross check the Table and Figure numbers given in the manuscript along with those in the main Table and figure files.

5. In the supporting tables, there are a number of abbreviations in each Table. Therefore, the authors are instructed to give the full forms of acronyms used along with the units of measurement in each supplementary tables so that the prospective readers do not have any difficulty in understanding.

6. In the discussion section of this study, I located some sentences which have already been summarized in the results part. For example, see the lines 330-332 as well as 345-349 which are indeed a repetition. Therefore, the author is advised to remove or modify them to avoid duplicacy.

7. A major finding of this study is that some polyembryonic lines like CSJ-8 and 14 are being proposed for use as salt tolerant rootstocks. The issue is that ‘are these materials being maintained in a field repository?’ Again, what would be the method of multiplication?

8. It has to be ensured that the references have been laid out as per journal style.

Reviewer 3 ·

Basic reporting

The authors have written the article in a coherent manner. The language of the paper is overall good and inclusive. Every important details of the experiment including the material used, plant growth conditions and salt treatment, and methodologies followed to determine various parameters are properly described. Data have been properly analyzed using different statistical tests. The results have been presented in a befitting manner along with proper discussion.

Experimental design

The methods and procedures described for the study are sufficient and reproducible. Proper experimental design has been used and choice of statistical tests is appropriate.

Validity of the findings

Authors have studied a fairly large number of genotypes, and assessed the effects of salinity using a wide range of parameters. The fact that selected genotypes differing in salt tolerance were also evaluated for major physiological parameters also improves the quality of results. Broadly speaking, the results presented and discussed in the paper justify the study objectives with some potential future applications.

Additional comments

In my opinion, the present study entitled “Similar and divergent responses to salinity stress of jamun (Syzygiumcumini L. Skeels) genotypes: biomass allocation and ion partitioning” is an important contribution to the field of salinity research in jamun; an underutilized fruit crop with significant medicinal benefits. As authors mention, jamun has so far attracted little research focus including the evaluation for salt tolerance. Therefore, I think that these experimental results are interesting and worthy for publication. However, the authors should be advised for some major changes before the manuscript is accepted. I am highlighting below my opinions and comments for the needful on the part of authors:

1. As authors have also assessed some additional observations such as gas exchange and anti oxidant enzymes, I suggest to keep title as ‘Similar and divergent responses to salinity stress of jamun (Syzygiumcumini L. Skeels) genotypes’ only.

2. There are some grammatical mistakes in the abstract which should be rectified.

3. In line 30-32 of abstract, the author mentions that ‘Leaf K+ was particularly little affected in genotypes with high leaf Na+, which likely ensured sequestration of excess Na+ in vacuoles’. I recommend to keep only first part of sentence ‘Leaf K+ was particularly little affected in genotypes with high leaf Na+’ and delete the remaining.

4. The line 67-68 of Introduction written as ‘Delineating such genetic differences may pave the way for their use as salt-tolerant scion and rootstock cultivars’ should be revised for a better meaning. For example as ‘Identifying such genetic differences may lead to the use of salt-tolerant genotypes as rootstocks for salt-affected soils’.

5. In the Materials part of the study, mention of the fact that treatments were stopped with the appearance of leaf injury symptoms in some genotypes is incomplete without the name of genotypes which should be clearly mentioned.

6. Authors have recorded gas exchange parameters and antioxidant enzyme in some genotypes; however they have not mentioned which leaf they measured for these attributes (new or mature). Same is the case for mineral ion analysis. So, it should be clearly stated in these sections.

7. In the statistical part, mention of ‘independent and interaction effects of salinity (fixed factor) and genotype (random factor) on the variance in different traits were assessed by a two-way ANOVA’ is not supported by suitable reference from which authors have drawn insight.

8. Parenthesis is not there after CSJ-21 (line 162). It should be added.

9. In line 169-175 of the results, I think there is scope for further revision and improvement and author should make necessary changes. This is due to fact that instead of separately for mono and polyembryonic types, these findings may be presented in continuity.

10. In lines 324-326 of Discussion, it is remarked that a sizeable number of genotypes need to be evaluated to accurately assess the variability for salt tolerance. However, word ‘accurately’ seems inappropriate here and may be replaced by ‘reasonably’.

11. In the discussion part, two previous findings have been cited (Patil & Patil, 1984 Yousaf et al. 2020). However, as the salinity levels in these studies were fairly similar (10-12 ds/m), they should be presented together and not separately. Thispresented together and not separately. This is also because both studies deal with effects of salinity on nutrient ions.

12. Authors should cross check all the references cited in the manuscript along with those given under references to ensure there are no discrepancies

---

## Round 0.2 · accepted · Accept

The authors have incorporated the suggestions and improved the manuscript. It can be accepted for publication.

·

Basic reporting

The authors have thoroughly revised the manuscript and addressed all the necessary changes in manuscript. Overall, the manuscript seems to be very good and might be accepted for publication in its present form.

Experimental design

No comments, in my opinion experimental design is okay.

Validity of the findings

Findings were very good and could help researchers and farmers to grow and study salt tolerant Jamun.

Additional comments

1. The findings improve the existing knowledge with respect to salt tolerance in jamun which is a neglected crop yet has a lot of therapeutic value.
2. The experimental results are interesting and worthy for publication.
3. Comparative assessment of salt tolerance of two seedling types (mono vs polyembryonic) is a major aspect of this study.

Reviewer 3 ·

Basic reporting

I have gone through the revised manuscript, and checked the responses listed in the rebuttal letter. I have seen that the author has made desired changes as per suggestions. Now, the manuscript is in order and in a easily understandable language. All the methods are properly described, and therefore the prospective researchers may use them in future studies. The results have been presented and discussed in a clear and coherent manner. The article is laid out as per journal format and sufficient literature has been cited and discussed.

Experimental design

The experimental materials and procedures are now adequately presented in the revised manuscript. The hypothesis of the study is well drafted. Authors have carried out a long-term investigation using various key plant growth and physiological attributes to derive logical conclusions.

Validity of the findings

As authors themselves state that there is paucity of information on salt tolerance of jamun. Therefore, the present investigation deserves merit. Authors have evaluated good number of genotypes from different parts of India. Statistical methods are sufficient and up to the expectation. The conclusions given in the paper are linked to the research hypothesis.

Additional comments

The manuscript may be accepted for publication.